# Lifecourse body mass index trajectories and cardio-metabolic disease risk in Guatemalan adults

**Nicole D. Ford**[1], **Reynaldo Martorell**[1], **Neil K. Mehta**[2], **Cria G. Perrine**[3], **Manuel Ramirez-Zea**[4], **Aryeh D. Stein**[1]*

**1** Hubert Department of Global Health, Rollins School of Public Health, Emory University, Atlanta, Georgia, United States of America, **2** Department of Health Management and Policy, School of Public Health, University of Michigan, Ann Arbor, Michigan, United States of America, **3** Division of Nutrition, Physical Activity, and Obesity, U.S. Centers for Disease Control and Prevention, Atlanta, Georgia, United States of America, **4** INCAP Research Center for the Prevention of Chronic Diseases (CIIPEC), Institute of Nutrition of Central America and Panama, Guatemala City, Guatemala

* aryeh.stein@emory.edu

**Data Availability Statement:** There are ethical or legal restrictions on sharing a de-identified data set. We cannot anonymize the data from this cohort as all individuals come from one of four previously

## Abstract

Little is known about body size over the life-course and non-communicable disease risk in low- and middle-income country populations. Our study explored the role of body mass index (BMI) trajectories from infancy through mid-adulthood on cardio-metabolic disease (CMD) risk factors in a prospective cohort of Guatemalan adults. Study participants were born in Guatemala from 1962–77 and have been followed prospectively since participating in a nutrition supplementation trial as children. Sex-specific BMI latent class trajectories were derived using latent class growth modeling from up to 22 possible BMI values from age 1 month to 42 years measured between 1969 and 2004. CMD risk factors were assessed in 2015–17 (at age 37–54 years) using anthropometry, blood glucose and lipids, and blood pressure. We used logistic regression to assess the role of BMI trajectory on CMD risk factors in 510 women and 346 men (N = 856). We identified two BMI latent classes for women (low [$n = 287$, 56.3%] and high [$n = 223$, 43.7%]) and three classes for men (low [$n = 141$, 40.8%], medium [$n = 160$, 46.2%], and high [$n = 45$, 13.0%]). Given the small percentage of men in the high BMI latent class, we collapsed the medium and high BMI latent classes for men ($n = 205$, 59.1%). Among the most prevalent CMD risk factors at ages 37–54 years were abdominal obesity defined by waist-height ratio (99.6% of women and 87.3% of men), obesity defined by percent body fat (96.6% of women and 75.9% of men), low HDL-c (87.5% of women and 74.5% of men), and elevated triglycerides (78.3% of women and 73.6% of men). Except for obesity defined by BMI, we found no associations between BMI latent class and CMD risk factors in women. Among men, BMI latent class was not associated with CMD risk factors after controlling for current BMI. For the CMD risk factors we analyzed, the role of early life BMI on adult CMD appeared to be mediated by adult BMI among men–highlighting the need to establish and maintain healthy body weight over the life course.

named villages and hence are readily re-identifiable once their demographic characteristics are known. We will not post data to a public archive, but we will make a replication data set available to bona fide researchers who agree to sign an LDUA and are covered under an IRB. Please contact the Research Center for the Prevention of Chronic Diseases (CIIPEC) at the Institute of Nutrition of Central America and Panama for requests. The local data protection manager is Dina Roche (email: droche@incap.int; phone: +502 5499 7220). SAS code for this project has been uploaded to the Open Science Framework and is available from: osf.io/5cvw8.

**Funding:** This work was supported by the National Institutes of Health (www.nih.gov; grant number HD-075784) and Laney Graduate School at Emory University (www.gs.emory.edu; to NDF). The funders had no role in: the design and conduct of this study; collection, management, analysis and interpretation of the data; preparation, review, or approval of the manuscript; and the decision to submit the manuscript for publication. The findings and conclusions in this report are those of the authors and do not necessarily represent the official position of the Centers for Disease Control and Prevention.

**Competing interests:** The authors have declared that no competing interests exist.

## Introduction

Obesity-related non-communicable diseases represent a growing proportion of the disease burden in Latin America [1] and are expected to double in the next 40 years [2]. Guatemala has a high burden of these conditions; in 2015, 67.2% of adults aged 18 years and older residing in Metropolitan Guatemala were overweight, 28.8% were obese, 28.4% had impaired fasting glucose, and 26.7% had elevated blood pressure [3]. According to the Global Burden of Disease study, high body mass index (BMI), high fasting plasma glucose, and high blood pressure are among the top contributors to combined death and disability [4].

Despite the rise in obesity prevalence, childhood undernutrition remains a serious problem in Guatemala. As of 2017, 42.6% of children aged under five years were stunted (height-for-age Z scores < -2 SD) while only 4.0% had overweight or obesity (BMI > 2 SD) [5]. Early life undernutrition is thought to increase risk of adult cardio-metabolic disease (CMD) [6]. Birth size has been consistently associated with adiposity, obesity, type 2 diabetes, lipid profile, and blood pressure later in childhood, adolescence, or adulthood [7–11]. The Developmental Origins of Health and Disease Hypothesis posits that structural and functional adaptations to early life undernutrition might preserve brain and vital organ development in contexts of nutrient deprivation; however, these adaptations are a "mismatch" for later obesogenic environments in countries undergoing nutrition transition [6]. Yet there is little data from low- and middle-income country (LMIC) populations on the long term consequences of persistent early life undernutrition in obesogenic environments [12].

Studies of life course growth trajectories and CMD risk factors have focused on high-income country populations [13–16]; however, growth patterns in LMIC are likely distinct due to persistent childhood undernutrition and differences in diet and physical activity. Studies from LMIC populations have explored the role of BMI in smaller segments of the life course, for example, adolescence, with CMD risk factors [10, 17–21]–the limited focus likely owing to the relatively young age of most LMIC birth cohorts [22].

Understanding the role of childhood/adolescent body size on adult disease risk is challenging for several reasons: adult BMI is correlated with both childhood BMI and adult CMD [23]; conventional mediation models do not work well with three or more repeated measures [24]; and growth models using a single curve to describe average growth of a population could obscure heterogeneity in growth patterns among sub-groups within that population [25]. In contrast, latent class growth analysis (LCGA) allows us to identify distinct growth patterns in cohort sub-groups not readily identifiable using other modeling techniques and to minimize collinearity of repeated measures [26]. If people exposed to low nutrient environments in early life–especially those later exposed to hyper-caloric environments consistent with developing economies–have growth patterns that confer distinct risk for CMD, life course growth analysis could help identify differential risk.

The objective of this research was to examine the role of life course BMI trajectories from infancy or early childhood through mid-adulthood on CMD risk factors in a prospective cohort of Guatemalan adults.

## Methods

### Study population

Study participants were born in four villages in southeastern Guatemala from 1962–77 and participated in the Institute of Nutrition of Central America and Panama (INCAP) Oriente Longitudinal Study (1969–77) and its follow-up studies (1989–2017) [27]. The original community-randomized intervention trial was designed to assess the influence of improved early

life nutrition on cognitive and physical development (N = 2,392). Two sets of matched villages were randomized to *Atole*, a protein-energy nutritional supplement, or *Fresco*, a low-energy beverage made from sugar and water. Children could be exposed prenatally through maternal consumption as well as postnatally through breastmilk or the child's own supplement intake. Exposure to *Atole* before age 3 years was positively with weight and linear growth and negatively associated with fat-folds, suggesting that increases in weight were largely due to fat-free mass [28]. The original trial provided the anthropometric measures in childhood and four follow-up waves (1988–89, 1997–99, 2002–04) provided measures in adolescence/adulthood for derivation of life course BMI trajectories. The 2015–17 follow-up (henceforth referred to as 2016) study provided anthropometric, biochemical, clinical, and sociodemographic information. Full details of the original trial and its follow-up studies are published elsewhere [29].

All data collection followed protocols that were approved by the institutional review boards of INCAP (Guatemala City, Guatemala) and Emory University (Atlanta, Georgia, United States of America). Caregivers consented on behalf of their children in the original trial (1969–77) and in the first follow-up wave (1988–89) if the participants were minors. For all other follow-up waves, participants gave written informed consent.

## Anthropometric, biochemical, and clinical measures

In the original trial and the four subsequent follow-up study waves, trained personnel collected length/height and weight data using standard procedures [30]. BMI was calculated as weight (kg) divided by height squared (m$^2$).

For CMD risk profiles in 2016, we assessed two measures of overall adiposity–BMI category and percent body fat–and two measures of central adiposity–abdominal obesity defined by waist circumference and by waist-height ratio (WHtR). We additionally evaluated blood lipids, fasting blood glucose, 2-hour post-challenge glucose, blood pressure, and metabolic syndrome. Metabolic syndrome (MetS) is a group of interrelated metabolic risk factors that increases risk of conditions such as cardiovascular disease, stroke, and diabetes [31].

In 2016, height and waist circumference were measured to the nearest 0.1 cm and weight to the nearest 0.01 kg. All measurements were taken in duplicate; if the difference exceeded 0.5 cm for height, 1.0 cm for waist circumference, or 0.5 kg for weight, a third measurement was taken and the average of the two closest measurements was used. To obtain percent body fat, we calculated total body water using the deuterium oxide dilution technique [32]. We estimated fat-free mass from total body water assuming that fat-free mass has a hydration constant of 0.732 and then subtracted fat-free mass from body mass to estimate fat mass. BMI was classified as: underweight/normal (<25.0 kg/m$^2$), overweight (25.0–29.9 kg/m$^2$), and obesity (≥30.0 kg/m$^2$) [33]. Obesity by percent body fat was defined as body fat ≥32% for women and ≥25% for men [34]. Abdominal obesity was defined as waist circumference >88 cm for women and >102 cm for men [33]. Waist-height ratio (WHtR) was calculated as waist circumference (cm) divided by height (cm). Abdominal obesity by WHtR was defined as WHtR >0.50 [35].

Trained phlebotomists drew venous blood samples in the fasted state (>8 hours) and 120 minutes after a prandial challenge. Lipids (total cholesterol, HDL-c, and triglycerides) and glucose concentrations were measured by enzymatic colorimetric methods (Cobas C111 analyzer, ROCHE, Indiana, United States of America). Elevated triglycerides were defined as triglycerides ≥150 mg/dL [36]. Low HDL-c was defined as <50 mg/dL for women and <40 mg/dL for men [36]. Prediabetes was classified as fasting plasma glucose 100–125 mg/dL or post-challenge glucose 140–199 mg/dL, and diabetes was classified as fasting plasma glucose ≥126 mg/dL, post-challenge glucose ≥200 mg/dL, or use of diabetes medication [37].

Seated blood pressure was measured three times at three-minute intervals on the left arm resting on a table at heart level using a digital blood pressure monitor (Omron, Schaumburg, Illinois, United States of America) after a five minute rest [38]. If systolic or diastolic blood pressure measurements differed by >10 mmHg, then a fourth measure was taken and the average of the two closest measurement was used; otherwise, the average of the second and third measurements was used. Pre-hypertension was defined as systolic blood pressure 120–129 mmHg and diastolic blood pressure <80 mmHg among participants without anti-hypertensive medication use [35]. Hypertension was defined as systolic blood pressure ≥130 mmHg and/or diastolic blood pressure ≥80 mmHg and/or anti-hypertensive medication use [39].

We defined MetS based on presence of ≥3 of the following: abdominal obesity (waist circumference >88 cm for women; >102 cm for men); fasting glucose ≥100 mg/dL or diabetes medication use; triglycerides ≥150 mg/dL or statin use; HDL-c <40 mg/dL in men or <50 mg/dL in women; and blood pressure ≥130 mmHg systolic, ≥85 mmHg diastolic, and/or hypertension medication use [31].

## Lifestyle and socioeconomic characteristics

We identified the following potential covariates based on their theoretical relationship with CMD risk factors including, socioeconomic status (SES), current residence, parity, smoking status, current vitamin intake, current alcohol consumption, and physical activity level. Data on lifestyle and socioeconomic factors in 2016 were collected by interview. Socioeconomic status in 2016 was a cumulative score developed from principal components analysis of household characteristics and consumer durable goods for participant households [40]. Current residence was classified as Guatemala City vs. other (rural and semi-rural). Parity was included as a continuous variable. Smoking status was classified as ever vs. never smoker. Current multivitamin and alcohol use were classified as yes or no. Physical activity level was ascertained using the International Physical Activity Questionnaire short form which has been validated for use in Guatemala [41]. We calculated participant physical activity level using the scoring protocol; participants who did not meet the criteria for moderate (any one of the following criteria: ≥3 days of vigorous activity of ≥20 min/day; or ≥5 days of moderate-intensity activity or walking of ≥30 min/day; or ≥5 days of any combination of walking, moderate-intensity or vigorous intensity activities achieving ≥600 MET-min/week) or high (any of the following criteria: vigorous-intensity activity on ≥3 days and accumulating ≥1,500 MET-minutes/week; or ≥7 days of any combination of walking, moderate-intensity or vigorous intensity activities achieving ≥3,000 MET-minutes/week) physical activity were classified as physically inactive [42].

## BMI trajectories

In previously published analyses [43], we derived BMI latent class trajectories from participant length/height and weight data in 1969–77, 1988–89, 1997–99, and 2002–04. Using LCGA, sex-specific BMI latent class trajectories were derived from up to 22 possible measures of height and weight: five from 1–12 months; three from 13–23 months; five from 24–50 months; three from 51–84 months; two from 10–20 years; and four from 21–42 years. Because a minimum of three BMI values improve model stability in LCGA [44], trajectories were derived for participants with ≥2 BMI values in childhood (0–84 months) from 1969–77 and ≥1 non-pregnant BMI value in adolescence/adulthood (10–42 years) from 1988–2004. Among those included, 5% had 3 measurements, 39% had 4–9 measurements, 34% had 10–14 measurements, and 23% had ≥15 measurements. Models were developed using all available data and robust maximum likelihood estimation, assessing overall model fit using the Bayesian Information

Criterion (BIC), the Bootstrap Likelihood Ratio (BLR) Test, and the Lo–Mendell–Rubin Likelihood Ratio (LMR) Test. The quality of classification was based on the entropy statistic and posterior probabilities [25, 44]. We examined 2-, 3-, and 4-class models for men and women.

Two BMI latent classes were identified for women: low ($n$ = 287, 56.3%) and high ($n$ = 223, 43.7%); and three classes for men low ($n$ = 141, 40.8%), medium ($n$ = 160, 46.2%), and high ($n$ = 45, 13.0%) (Fig 1). The 2-class model for women had the highest entropy (0.76) and had significant LMR and BLR tests ($P$ = 0.008 and $P<0.001$, respectively). Among men, the 3-class model had the lowest BIC and the highest entropy (0.77) and had significant LMR and BLR tests ($P<0.001$ for both). Given the small percentage of men in the high BMI latent class leading to non-converging models in the multivariable analyses, we collapsed the medium and high BMI latent classes for men ($n$ = 205, 59.1%).

## Statistical analyses

Of the 2,392 participants in the original study, 369 (15.4%) of the participants in the original trial had died (the majority in early childhood), 249 (10.4%) had migrated outside of Guatemala, and 113 (4.7%) were untraceable, and 1,661 (69.4%) were presumed alive and living in Guatemala as of 2015 (Fig 2). Of the 1,661 participants eligible for enrollment, 1,161 (69.9%) provided informed consent. Of these, we excluded those who were pregnant/lactating in 2016 (n = 6), who did not attend the clinical examination (n = 16), who were missing BMI latent class trajectory due to insufficient number BMI values (n = 281). We also excluded two participants who were missing covariates. Those excluded were on average older, had higher parity (women), and were more likely to reside in Guatemala City (men) (data not presented). At least one outcome measure was obtained from 510 women and 346 men, ages 37–54 years.

To assess differences in sociodemographic and health characteristics across BMI latent class trajectories, we used chi-square tests for categorical variables and t tests for continuous variables. Among the potential covariates, we selected the final covariates by assessing bivariate relationships between the potential covariates and the outcomes. Parity, multivitamin intake, and alcohol consumption were not included in the final adjusted models. To characterize the relationship between BMI latent class trajectory and each CMD risk factor in 2016, we used sex-stratified logistic regression models to regress dichotomous CMD risk factor outcomes (obesity defined by BMI, obesity defined by percent body fat, abdominal obesity defined by waist circumference, abdominal obesity defined by WHtR, elevated triglycerides, low HDL-c, diabetes, hypertension, and MetS) on BMI latent class trajectory. Model 1 included BMI latent class trajectory, age at the time of clinical assessment, and birth village. Model 2 additionally controlled for current residence, SES, and lifestyle covariates (low physical activity level and smoking status) in 2016. To evaluate the mediating role of current BMI on CMD risk factors, Model 3 additionally controlled for BMI in 2016 in all models except where obesity defined by BMI was the outcome. Because many participants had at least one sibling in the study, we used generalized estimating equations to account for clustering at the mother level.

In sensitivity analyses, we evaluated the association between CMD risk factors in 2016 with: (1) a single measure of adiposity in childhood (weight-for-height Z score [WHZ] at age 18–42 months); and (2) a measure of adiposity in adolescence/young adulthood (BMI in 1988–89 at age 10–27 years) using the same modeling strategy described above for the primary analyses. Analyses included all non-pregnant/lactating participants for whom complete CMD risk factor information in 2016 and each respective adiposity value were available (n = 668 for WHZ analyses and n = 752 for 1988–89 BMI). We used the 2006 World Health Organization Multicentre Growth Reference child growth standards to calculate each child's WHZ, standardized to the reference population for the child's age and sex [45].

A

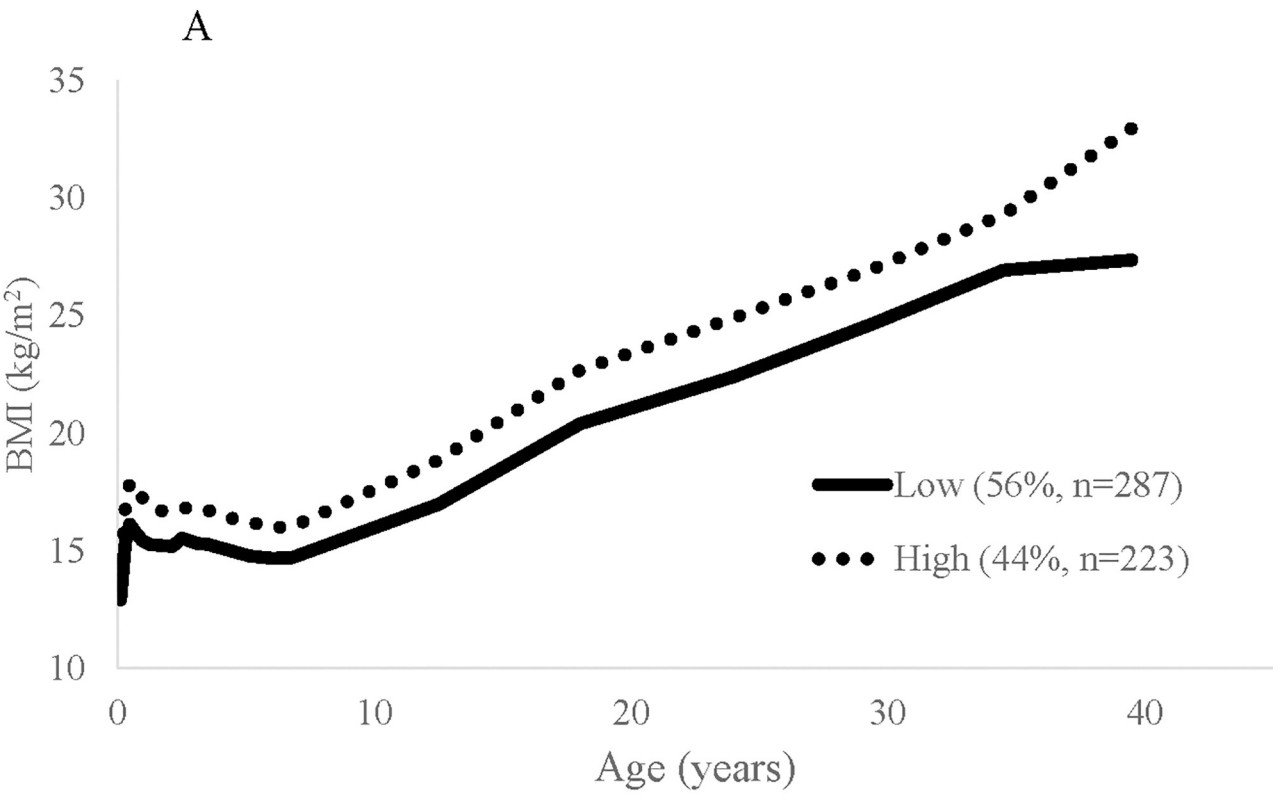

B

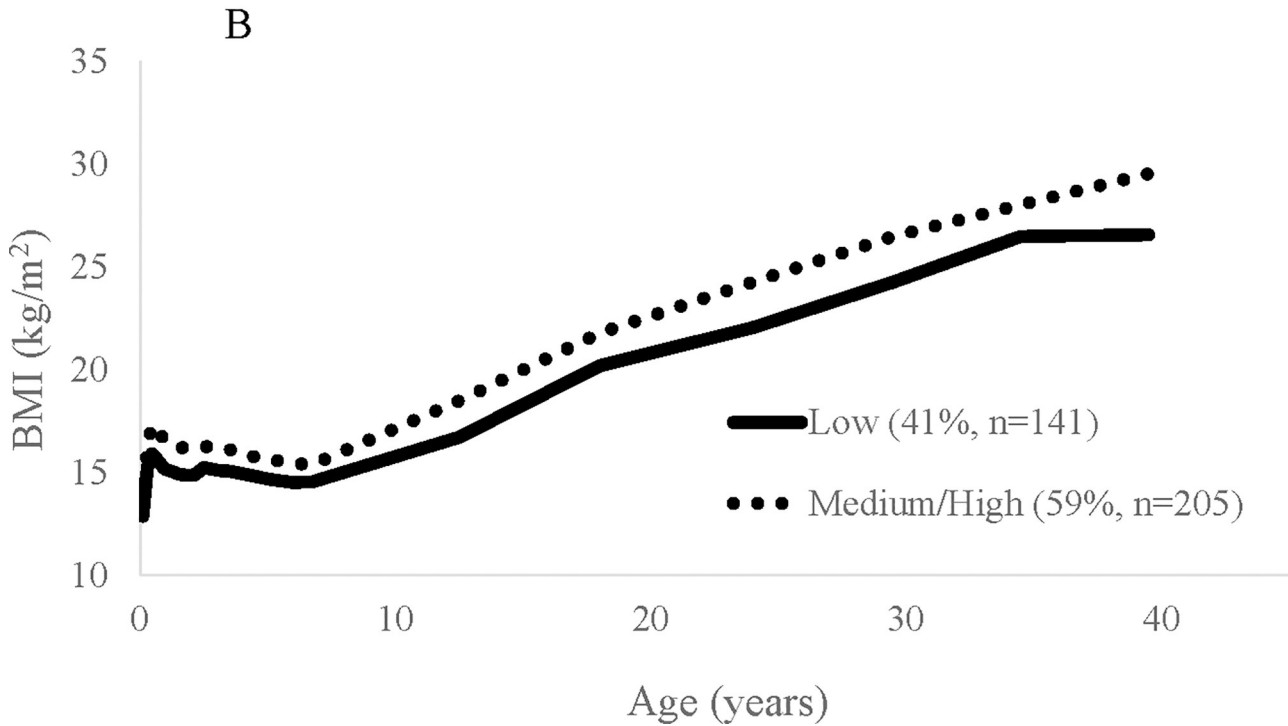

**Fig 1.** Mean body mass index (BMI) by BMI latent class trajectory group in females (A) and males (B) in the INCAP Nutrition Supplementation Trial Longitudinal Cohort. Sex-specific BMI latent class trajectories were derived from 22 possible measures of height and weight from 1969–2004 using latent class growth modeling.

Statistical significance was set *a priori* at $P < 0.05$. All *P*-values were two-sided. All analyses were performed using SAS v.9.4 (SAS Institute, Cary, North Carolina, United States of America). SAS code is available online via OSF at: https://osf.io/5cvw8/.

## Results

The study sample was 59.6% female (Table 1). Fifty-six percent of women and 40.8% of men were in the low BMI latent class trajectory. In 2016, the difference in median BMI between the low and high/medium BMI latent class trajectories was 2.0 kg/m$^2$ for women and 1.8 kg/m$^2$ for men. Higher BMI latent class trajectories were more likely to be comprised of individuals in the more recent birth cohorts ($P < 0.0001$ for both sexes), suggesting a secular trend of earlier onset of high BMI among younger cohort members.

Among the most prevalent risk factors were abdominal obesity defined by WHtR (99.6% of women and 87.3% of men), obesity defined by percent body fat (96.6% of women and 75.9% of men), low HDL-c (87.5% of women and 74.5% of men), elevated triglycerides (78.3% of women and 73.6% of men), and MetS (81.3% of women and 46.5% of men). In bivariate analyses, current BMI was highly associated with CMD risk factors ($P < 0.001$ for all comparisons; data not presented).

Relative to the low BMI latent class, high (women) and high/medium (men) BMI latent classes had higher unadjusted median BMI, WHtR, and unadjusted prevalence of obesity defined by BMI ($P < 0.05$ for all comparisons). Relative to men in the low BMI latent class, men in the high/medium BMI latent class also had higher unadjusted median waist circumference, percent body fat and number of MetS components, and higher unadjusted prevalence of abdominal obesity defined by waist circumference and by WHtR, obesity defined by percent body fat, elevated triglycerides, and low HDL-c ($P < 0.05$ for all comparisons).

Among women, high BMI latent class was positively associated with obesity defined by BMI (adjusted Odds Ratio [aOR] 2.39, 95% CI 1.58, 3.60) relative to low BMI latent class in models controlling for age, birth village, residence, SES, and lifestyle factors (Table 2). Among men, high/medium BMI latent class was positively associated with obesity defined by BMI (aOR 2.35, 95% CI 1.21, 4.55), abdominal obesity (aOR 2.35, 95% CI 1.25, 4.42), obesity defined by percent body fat (aOR 2.04, 95% CI 1.15, 3.62), elevated triglycerides (aOR 1.81, 95% CI 1.06, 3.06), and low HDL-c (aOR 1.93, 95% CI 1.09, 3.40) relative to the low BMI latent class in fully adjusted models without BMI in 2016 (Model 2); however, these associations were not significant after adjusting for current BMI (Model 3).

In the mediation analyses, BMI in 2016 was strongly associated with all CMD risk factor outcomes in both sexes except for diabetes (S1 Table).

In sensitivity analyses, findings for models using WHZ at age 18–42 months as the exposure and models using BMI in 1988–89 at age 10–27 years as the exposure were consistent with those in the primary analyses (S2 and S3 Tables). Higher WHZ and 1988–89 BMI were generally associated with increased odds of CMD risk factors; however, apart from diabetes, these associations were not significant after adjusting for current BMI.

## Discussion

Using data from a longitudinal cohort with >40 years of follow-up, we examined the role of BMI latent class trajectories from infancy through mid-adulthood on CMD risk factors in

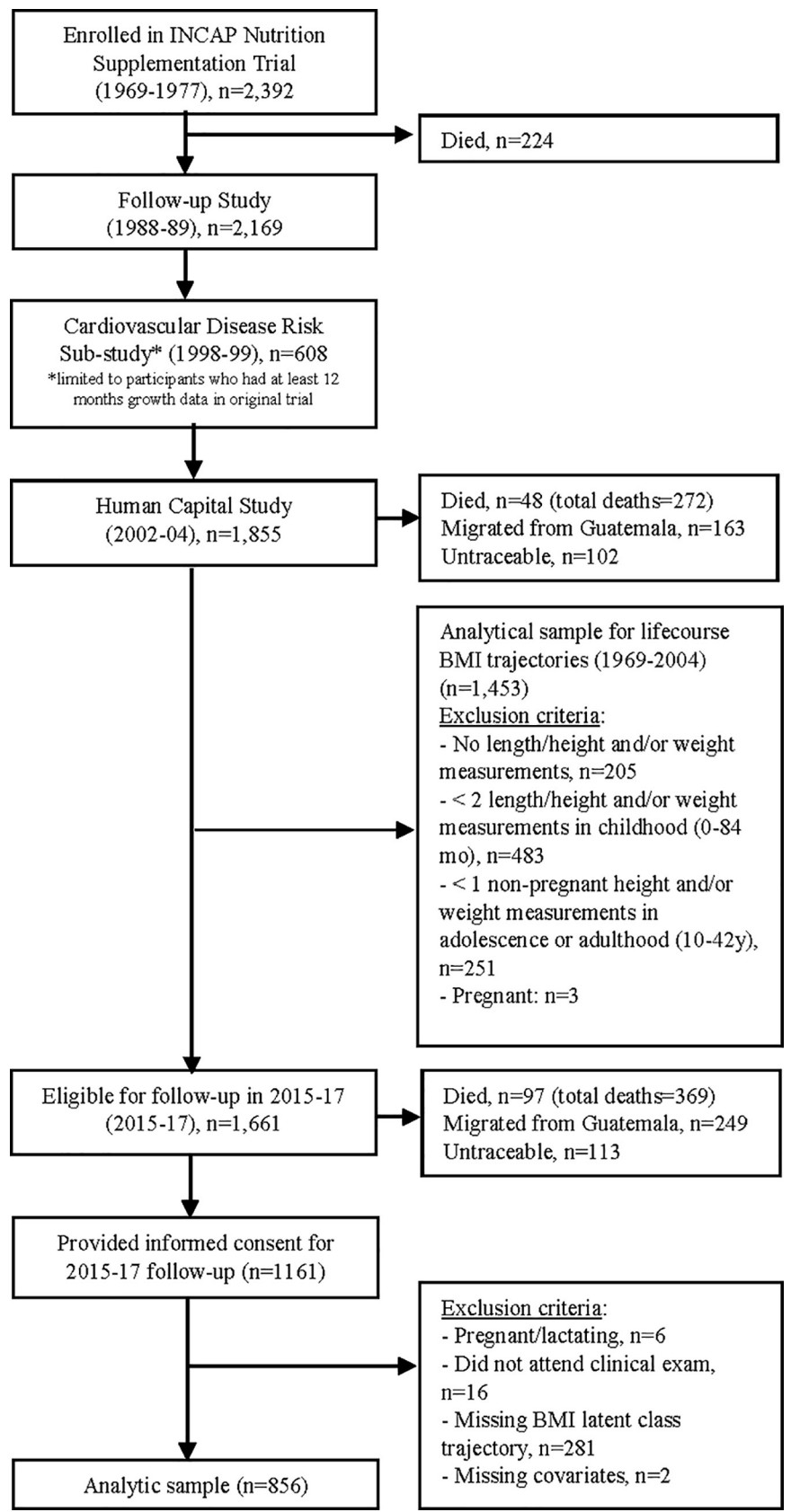

**Fig 2. Tracking of the analytic sample.** In 2015, of the original 2,392 individuals in the 1969–77 INCAP Nutrition Supplementation Trial, 15.4% (n = 369) had died, 10.4% (n = 249) had migrated from Guatemala, and 4.7% (n = 113) of the original cohort members were untraceable. Of the 1,661 original cohort members eligible for the 2015–17 follow-up study, 1,161 provided informed consent.

Guatemalan adults. Our work previously identified two BMI latent classes for women (low and high) and three classes for men, which we have collapsed into two (low and medium/high) [43]. Except for obesity defined by BMI, we found no associations between BMI latent class and CMD risk factors among women in 2016 –possibly due to high levels of adiposity in both trajectory classes. Among men, high/medium BMI latent class was positively associated with obesity (defined by BMI, waist circumference, and percent body fat), elevated triglycerides, and low HDL-c; however, associations appeared to be mediated by current BMI. In sensitivity

**Table 1. Select sociodemographic and health characteristics in 2015–17 after 40 years of follow up at age 37–54 years by BMI latent class trajectory and sex, INCAP nutrition supplementation trial longitudinal cohort (n = 510 women, n = 346 men).**

| Characteristics | Women | | | | | Men | | | | |
| --- | --- | --- | --- | --- | --- | --- | --- | --- | --- | --- |
| | Low Class (n = 287) | | High Class (n = 223) | | | Low Class (n = 141) | | Medium/High Class (n = 205) | | |
| | n | n (%) or Median (Q1, Q3) | n | n (%) or Median (Q1, Q3) | P-value[a] | n | n (%) or Median (Q1, Q3) | n | n (%) or Median (Q1, Q3) | P-value[a] |
| Age, years | 287 | 46.0 (42.0,49.0) | 223 | 42.0 (40.0,44.0) | <0.0001 | 141 | 45.0 (43.0,48.0) | 205 | 42.0 (40.0,46.0) | <0.0001 |
| Reside in Guatemala City, % | 287 | 42 (14.6%) | 223 | 49 (22.0%) | 0.03 | 141 | 21 (14.9%) | 205 | 35 (17.1) | 0.6 |
| SES tertile, % | 287 | | 223 | | 0.4 | 141 | | 205 | | 0.7 |
| Poorest | | 91 (31.7%) | | 77 (34.5%) | | | 50 (35.5%) | | 65 (31.7%) | |
| Middle | | 107 (37.3%) | | 71 (31.8%) | | | 44 (31.2%) | | 63 (30.7%) | |
| Wealthiest | | 89 (31.0%) | | 75 (33.6%) | | | 47 (33.3%) | | 77 (37.6%) | |
| Parity, n | 287 | 3.0 (2.0,4.0) | 223 | 3.0 (2.0,5.0) | 0.5 | - | - | - | - | - |
| Alcohol use, % | 287 | 13 (4.5%) | 223 | 10 (4.5%) | 0.9 | 141 | 61 (43.3%) | 205 | 71 (34.6%) | 0.1 |
| Ever smoker, % | 287 | 11 (3.8%) | 223 | 9 (4.0%) | 0.9 | 141 | 89 (63.1%) | 205 | 130 (63.4%) | 0.9 |
| Low physical activity[b], % | 287 | 149 (51.9%) | 223 | 127 (57.0%) | 0.2 | 141 | 66 (46.8%) | 205 | 92 (44.9%) | 0.7 |
| Multivitamin use, % | 287 | 38 (13.2%) | 223 | 24 (10.8%) | 0.4 | 141 | 26 (18.4%) | 205 | 18 (8.8%) | 0.008 |
| BMI, kg/m$^2$ | 287 | 28.0 (25.5,31.3) | 223 | 30.0 (26.4,33.4) | 0.0002 | 141 | 25.3 (22.8,28.1) | 205 | 27.1 (25.1,29.8) | <0.0001 |
| Obesity defined by BMI[c], % | 287 | 94 (32.8%) | 223 | 112 (50.2%) | <0.0001 | 141 | 17 (12.1%) | 205 | 47 (22.9%) | 0.01 |
| Waist circumference, cm | 287 | 99.2 (93.0,107.7) | 223 | 102.5 (95.0,110.1) | 0.06 | 141 | 92.5 (85.1,98.6) | 205 | 95.0 (89.7,102.2) | 0.0008 |
| Abdominal obesity defined by waist circumference[d], % | 287 | 257 (89.5%) | 223 | 203 (91.0%) | 0.6 | 141 | 20 (14.2%) | 205 | 52 (25.4%) | 0.01 |
| Waist-height ratio | 287 | 0.65 (0.61,0.71) | 223 | 0.68 (0.62,0.73) | 0.03 | 141 | 0.56 (0.52,0.59) | 205 | 0.59 (0.55,0.62) | <0.0001 |
| Abdominal obesity defined by waist-height ratio[e], % | 287 | 286 (99.7%) | 223 | 222 (99.6%) | 0.8[f] | 141 | 113 (80.1%) | 205 | 189 (92.2%) | 0.0009 |
| Body fat, % | 279 | 42.4 (37.9,45.7) | 218 | 42.8 (39.4,46.5) | 0.3 | 133 | 28.5 (23.9,33.2) | 199 | 29.6 (25.8,34.3) | 0.04 |
| Obesity defined by % body fat[g], % | 279 | 268 (96.1%) | 218 | 212 (97.2%) | 0.5 | 133 | 93 (69.9%) | 199 | 159 (79.9%) | 0.02 |
| Elevated triglycerides[h], % | 284 | 219 (77.1%) | 219 | 175 (79.9%) | 0.4 | 136 | 91 (66.9%) | 197 | 154 (78.2%) | 0.02 |
| Low HDL-c[i], % | 284 | 243 (85.6%) | 219 | 197 (90.0%) | 0.1 | 136 | 93 (68.4%) | 197 | 155 (78.7%) | 0.03 |
| Blood pressure | 287 | | 223 | | 0.2 | 141 | | 205 | | 0.5 |
| Pre-hypertension[j], % | | 47 (16.4%) | | 39 (17.5%) | | | 41 (29.1%) | | 41 (20.0%) | |
| Hypertension[k], % | | 122 (42.5%) | | 81 (36.3%) | | | 45 (31.9%) | | 76 (37.1%) | |
| Dysglycemia | 285 | | 219 | | 0.01 | 136 | | 197 | | 0.4 |
| Prediabetes[l], % | | 112 (39.3%) | | 64 (29.2%) | | | 51 (37.5%) | | 57 (28.9%) | |
| Diabetes[m], % | | 47 (16.5%) | | 34 (15.5%) | | | 11 (8.1%) | | 24 (12.2%) | |
| Metabolic syndrome risk factors | 284 | 3.0 (3.0,4.0) | 219 | 3.0 (3.0,4.0) | 0.4 | 136 | 2.0 (1.0,3.0) | 197 | 2.0 (2.0,4.0) | 0.01 |

*(Continued)*

**Table 1.** (Continued)

| Characteristics | Women | | | | | Men | | | | |
|---|---|---|---|---|---|---|---|---|---|---|
| | Low Class (n = 287) | | High Class (n = 223) | | | Low Class (n = 141) | | Medium/High Class (n = 205) | | |
| Metabolic syndrome[n], % | 284 | 232 (81.7%) | 219 | 177 (80.8%) | 0.8 | 136 | 58 (42.6%) | 197 | 97 (49.2%) | 0.2 |

Values presented are median (Q1, Q3) or percentages. Sex-specific BMI latent class trajectories were derived using latent class growth modeling from up to 22 BMI values from age 1 month to 42y measured between 1969 and 2004.

a. *P* was calculated using t tests (continuous) and chi-square tests (categorical variables).

b. Low physical activity in 2015–17 defined as participants who do not meet the International Physical Activity Questionnaire (IPAQ) scoring criteria for moderate or high physical activity.

c. Obesity by BMI defined as BMI $\geq$30 kg/m$^2$.

d. Abdominal obesity defined as waist circumference >88 cm for women and >102 cm for men.

e. Abdominal obesity by waist-height ratio defined as waist-height ratio >0.50.

f. Estimate unstable due to small cell sizes.

g. Obesity by percent body fat defined as body fat $\geq$32% for women and $\geq$25% for men.

h. Elevated triglycerides defined as $\geq$150 mg/dL or medication.

i. Low HDL-c defined as HDL-c <50 mg/dL for women and <40 mg/dL for men.

j. Pre-hypertension defined according to the 2017 ACC/AHA/AAPA/ABC/ACPM/AGS/APhA/ASH/ASPC/NMA/PCNA Guideline for the Prevention, Detection, Evaluation, and Management of High Blood Pressure in Adults: systolic blood pressure 120–129 mmHg and/or diastolic blood pressure <80 mmHg among participants without self-reported hypertension and/or anti-hypertensive medication use.

k. Hypertension defined according to the 2017 ACC/AHA/AAPA/ABC/ACPM/AGS/APhA/ASH/ASPC/NMA/PCNA Guideline for the Prevention, Detection, Evaluation, and Management of High Blood Pressure in Adults: systolic blood pressure $\geq$130 mmHg and/or diastolic blood pressure $\geq$90 mmHg and/or anti-hypertensive medication use.

l. Prediabetes defined according to the American Diabetes Association diagnostic criteria: impaired fasting glucose (fasting plasma glucose 100–125 mg/dL) and/or impaired glucose tolerance (post challenge glucose 140–199 mg/dL).

m. Diabetes defined according to the American Diabetes Association diagnostic criteria: fasting plasma glucose $\geq$126 mg/dL, and/or post-challenge glucose $\geq$200 mg/dL, and/or diabetes medication use.

n. Metabolic syndrome defined according to the American Heart Association/National Heart, Lung, and Blood Institute scientific statement diagnostic criteria based on presence $\geq$3 of the following: abdominal obesity (waist circumference >88 cm for women and >102 cm for men); fasting plasma glucose $\geq$100 mg/dL or medication; triglycerides $\geq$150 mg/dL or medication; HDL-c <50 mg/dL for women and <40 mg/dL for men; and blood pressure $\geq$130 mmHg systolic, $\geq$85 mmHg diastolic and/or medication use.

Abbreviations: BMI, body mass index; HDL-c, high density lipoprotein cholesterol; INCAP, Institute of Nutrition for Central America and Panama.

**Table 2. Multivariable logistic regression models to predict cardio-metabolic disease risk factors in 2015–17 after 40 years of follow up at age 37–54 years based on body mass index latent class trajectory from infancy through mid-adulthood (high vs. low in women and high/medium vs. low in men) in the INCAP nutrition supplementation trial longitudinal cohort (n = 510 women, n = 346 men).**

| Cardio-metabolic risk factor | Women High vs. low | | Men High/medium vs. low | |
|---|---|---|---|---|
| | Adjusted Odds Ratio (95% CI) | *P* | Adjusted Odds Ratio (95% CI) | *P* |
| Obesity defined by BMI[a] | | | | |
| Model 1 | 2.24 (1.50, 3.38) | <0.0001 | 2.53 (1.32, 4.83) | 0.005 |
| Model 2 | 2.39 (1.58, 3.60) | <0.0001 | 2.35 (1.21, 4.55) | 0.01 |
| Model 3 | - | - | - | - |
| Abdominal obesity defined by waist circumference[b] | | | | |
| Model 1 | 1.15 (0.56, 2.34) | 0.7 | 2.44 (1.31, 4.53) | 0.005 |
| Model 2 | 1.24 (0.59, 2.58) [c] | 0.6 | 2.35 (1.25, 4.42) | 0.008 |
| Model 3 | 0.73 (0.25, 2.08) [c] | 0.5 | 1.15 (0.33, 3.97) | 0.8 |
| Obesity defined by percent body fat[d] | | | | |
| Model 1 | 1.06 (0.29, 3.78) | 0.9 | 2.04 (1.18, 3.53) | 0.01 |
| Model 2 | 1.17 (0.29, 4.66) [c] | 0.8 | 2.04 (1.15, 3.62) | 0.01 |

(*Continued*)

**Table 2.** (Continued)

| Cardio-metabolic risk factor | | Women High vs. low | | Men High/medium vs. low | |
|---|---|---|---|---|---|
| | | Adjusted Odds Ratio (95% CI) | P | Adjusted Odds Ratio (95% CI) | P |
| | Model 3 | 0.83 (0.21, 3.29) [c] | 0.8 | 0.96 (0.48, 1.92) | 0.9 |
| Elevated triglycerides[e] | | | | | |
| | Model 1 | 1.30 (0.81, 2.07) | 0.3 | 1.92 (1.15, 3.21) | 0.01 |
| | Model 2 | 1.36 (0.84, 2.21) | 0.2 | 1.81 (1.06, 3.06) | 0.03 |
| | Model 3 | 1.21 (0.74, 1.98) | 0.4 | 1.25 (0.71, 2.21) | 0.4 |
| Low HDL-c[f] | | | | | |
| | Model 1 | 1.33 (0.74, 2.37) | 0.3 | 1.93 (1.13, 3.32) | 0.02 |
| | Model 2 | 1.61 (0.88, 2.94) [c] | 0.1 | 1.93 (1.09, 3.40) | 0.02 |
| | Model 3 | 1.33 (0.71, 2.47) [c] | 0.4 | 1.27 (0.70, 2.32) | 0.4 |
| Diabetes[g] | | | | | |
| | Model 1 | 1.49 (0.83, 2.66) | 0.2 | 2.20 (0.92, 5.26) | 0.08 |
| | Model 2 | 1.50 (0.84, 2.73) | 0.2 | 2.10 (0.88, 5.03) | 0.1 |
| | Model 3 | 1.50 (0.83, 2.73) | 0.2 | 2.09 (0.83, 5.25) | 0.1 |
| Hypertension[h] | | | | | |
| | Model 1 | 0.86 (0.57, 1.29) | 0.5 | 1.43 (0.88, 2.32) | 0.1 |
| | Model 2 | 0.83 (0.54, 1.25) | 0.4 | 1.41 (0.86, 2.30) | 0.2 |
| | Model 3 | 0.66 (0.43, 1.03) | 0.07 | 1.06 (0.63, 1.77) | 0.8 |
| Metabolic syndrome[i] | | | | | |
| | Model 1 | 1.15 (0.72, 1.84) | 0.5 | 1.55 (0.96, 2.52) | 0.07 |
| | Model 2 | 1.24 (0.77, 1.99) | 0.4 | 1.53 (0.94, 2.50) | 0.09 |
| | Model 3 | 0.92 (0.56, 1.51) | 0.7 | 0.79 (0.44, 1.42) | 0.4 |

Sample sizes were 510 and 346 (obesity defined by BMI, abdominal obesity defined by waist circumference, hypertension), 504 and 333 (elevated triglycerides, low HDL-c, metabolic syndrome), 504 and 333 (diabetes), and 497 and 332 (obesity defined by percent body fat) for women and men, respectively. Values are odds ratios and 95% confidence intervals for BMI latent class trajectory from infancy through mid-adulthood (high vs. low in women and high/medium vs. low in men) controlling for: age and birth village (Model 1); current residence, SES, low physical activity, and smoking status in 2015–17 (Model 2); and BMI in 2015–17 (Model 3). Confidence intervals account for clustering at the mother level. Sex-specific BMI latent class trajectories were derived using latent class growth modeling from up to 22 BMI values from age 1 month to 42y measured between 1969 and 2004.

a. Obesity by BMI defined as BMI $\geq$30 kg/m$^2$.

b. Abdominal obesity defined as waist circumference >88 for women and >102 cm for men.

c. Modeled without smoking status due to non-convergence.

d. Obesity by percent body fat defined as body fat $\geq$32% for women and $\geq$25% for men.

e. Elevated triglycerides defined as $\geq$150 mg/dL or statin use.

f. Low HDL-c defined as HDL-c <50 mg/dL for women and <40 mg/dL for men.

g. Diabetes defined according to the American Diabetes Association diagnostic criteria: fasting plasma glucose $\geq$126 mg/dL, and/or post-challenge glucose $\geq$200 mg/dL, and/or diabetes medication use.

h. Hypertension defined according to the 2017 ACC/AHA/AAPA/ABC/ACPM/AGS/APhA/ASH/ASPC/NMA/PCNA Guideline for the Prevention, Detection, Evaluation, and Management of High Blood Pressure in Adults: systolic blood pressure $\geq$130 mmHg and/or diastolic blood pressure $\geq$90 mmHg and/or anti-hypertensive medication use.

i. Metabolic syndrome defined according to the American Heart Association/National Heart, Lung, and Blood Institute scientific statement diagnostic criteria based on presence $\geq$3 of the following: abdominal obesity (waist circumference >88 cm for women and >102 cm for men); fasting plasma glucose $\geq$100 mg/dL or medication; triglycerides $\geq$150 mg/dL or medication; HDL-c <50 mg/dL for women and <40 mg/dL for men; and blood pressure $\geq$130 mmHg systolic, $\geq$85 mmHg diastolic and/or medication use.

Abbreviations: BMI, body mass index; HDL-c, high density lipoprotein cholesterol; INCAP, Institute of Nutrition for Central America and Panama; SES, socioeconomic status.

analyses, WHZ at age 18–42 months and BMI at age 10–27 years were both generally associated with increased odds of CMD risk factors. Because BMI tracked from early childhood into adulthood and current BMI was strongly associated with CMD risk factors, our findings highlight the importance of keeping a healthy BMI throughout life.

In this Guatemalan cohort, adult BMI was strongly correlated with childhood BMI and was highly associated with all CMD risk factors in bivariate analyses. Trajectory differences were established in early infancy and maintained throughout the life course, suggesting that pre-conceptual or early life factors may influence BMI trajectory. Nutrition supplementation from conception to age 2 years was not associated with BMI latent class membership; however, higher childhood socioeconomic status was associated with increased odds of high BMI latent class membership in both men and women [43]. Data from children in the United States has also shown that BMI tracks from childhood into adulthood [46], and tracking begins as early as six months of age [47, 48]. Some study participants were exposed to a nutrition supplement that improved child growth; however, 86% of the cohort was stunted at age 2 years (HAZ <-2 SD), suggesting severe early life undernutrition even with the intervention. Despite low levels of childhood overweight, none of the BMI latent class trajectory classes had a median BMI in the normal range in 2016.

Even though the BMI latent classes were not clinically distinct with respect to CMD risk factors after adjusting for current BMI, our study has potentially important implications for prevention of chronic disease risk in contexts with early life undernutrition and later life caloric excess. Irrespective of BMI from childhood through mid-adulthood, current BMI was strongly associated CMD risk factor odds in this population, except for diabetes. Thus, maintaining healthy weight throughout the life course, including averting weight gain in mid-life, could be important in mitigating adult CMD risk. Weight gain in early and mid-adulthood has been associated with increased risk of many chronic diseases, including diabetes, cardiovascular disease, and certain cancers [49]. In a prospective cohort in the United States, participants who maintained a stably lean body shape over the life course (age 5–50 years) had the lowest risk of mortality while people who were lean in early life but had marked increase in body weight in middle age had mortality risk similar to participants who had been heavy throughout life [50]. Among men in our study, the apparent benefit of lower BMI from childhood to early adulthood with respect to lipid profile was negated by weight gain between 2004 and 2016.

High levels of adiposity in adulthood might explain the lack of association between BMI latent class and CMD risk factors in women in this study. In 2016, 32.8% of women in the low BMI latent class and 50.2% of women in the high BMI latent class had obesity defined by BMI, and >89% of women in both BMI latent classes had obesity defined by percent body fat and had abdominal obesity. Further, median unadjusted percent body fat and waist circumference and the prevalence of abdominal obesity defined by waist circumference and by WHtR and obesity defined by body fat percentage were not statistically different across BMI latent classes, meaning women in the low BMI latent class trajectory group still largely had obesity despite having a lower BMI latent class trajectory and lower median BMI in 2016. Considering BMI > 23 kg/m$^2$ increases risk of CMD [51] and neither of the women's BMI latent classes had a median BMI in the normal range, high levels of and lack of variation in adiposity in women could explain our inability to detect associations between BMI latent class trajectory and CMD risk factors. However, to our knowledge, the INCAP longitudinal cohort is the only study population available to answer our research question, owing to the relatively young age of the study participants in other LMIC cohorts.

It is also possible that BMI may not capture the type of adiposity that is important for adult CMD risk factors in this cohort–especially considering weight varies in its type (i.e. organs,

adipose tissue, lean mass, etc.) and location on the body. The risk of developing diabetes rises with increases in BMI; however, within a narrow range of BMI levels, there is high inter-individual variation in systemic inflammation and insulin resistance, possibly attributable to differences in the distribution and type of body fat which are thought to have varied local and/or systemic effects on metabolic dysfunction [52]. A systematic review using data including Central American participants found that current WHtR independently and more strongly predicted diabetes than did BMI [35]. A meta-analysis of WHtR, BMI, and chronic disease had similar findings with respect to MetS [53]. In our study, among participants with BMI <30 kg/m$^2$, 94.2% of women and 71.5% of men had obesity defined by body fat percentage, 83.5% of women and 5.0% of men were considered to have obesity by waist circumference, and 99.3% of women and 84.4% of men were considered to have obesity by WHtR (S4 Table). The disparate classification of obesity by different methods (BMI vs. waist-based measures vs. directly assessed body fat), lifecourse trajectories of waist circumference or body fat might better characterize later life CMD risk factors above current BMI in this stunted population.

The INCAP cohort is distinctive in that its participants have been followed for >40 years and have serial, clinically-measured anthropometry and CMD risk factors. To our knowledge, we are the first study to explore the role of BMI trajectories from infancy through mid-adulthood and CMD risk factors in a LMIC population. The 12-year lag between when the final weight/height in the BMI latent class trajectory and the CMD risk factors attempted to control the influence of potential reverse causation between adult BMI and CMD risk. Our findings were robust in models using WHZ from 18–42 months and BMI in 1988–89 as the primary exposures in sensitivity analyses, likely because BMI tracked over time.

Life course analyses are threatened by bias owing to missing data; however, those excluded from analyses for missing data were not substantially different from those included in this study. Sensitivity analyses using a single measure of adiposity in childhood and in adolescence/young adulthood were consistent with the findings of the primary analyses. Prior work has indicated that attrition has not biased estimates of early life exposures and later-life outcomes [54]. While LCGA helps identify heterogeneity in body size over life course, the classes are not "real" but instead reflecting a continuum of growth in the population and should be considered a tool to help visualize variability within the global distribution of BMI gain [55]. The generalizability of our findings may be limited to contexts of high childhood undernutrition with subsequent exposure to obesogenic adult environments.

## Conclusions

There was a high burden of CMD risk factors in this cohort–particularly among women. There is a strong need for chronic disease risk factor prevention and management in this population. Because BMI latent class trajectory was not associated CMD risk factors after controlling for current BMI and given that early life BMI tracked into adulthood, our findings highlight the importance of healthy growth early in life and preventing weight gain in adulthood.

## Supporting information

**S1 Table. Effect sizes for BMI in 2015–17 after 40 years of follow up at age 37-54y from multivariable logistic regression models predicting cardio-metabolic disease risk factors in 2015–17 in the INCAP nutrition supplementation trial longitudinal cohort (n = 510 women, n = 346 men).**
(DOCX)

**S2 Table. Multivariable logistic regression models to predict cardio-metabolic risk factors in 2015–17 after 40 years of follow up at age 37-54y based on weight-for-height Z score (WHZ) at age 18–42 months in the INCAP nutrition supplementation trial longitudinal cohort (n = 387 women, n = 281 men).**
(DOCX)

**S3 Table. Multivariable logistic regression models to predict cardio-metabolic risk factors in 2015–17 after 40 years of follow up at age 37-54y based on BMI in 1988–89 in the INCAP nutrition supplementation trial longitudinal cohort (n = 460 women, n = 292 men).**
(DOCX)

**S4 Table. Classifications of obesity defined by percent body fat, waist circumference, and waist-height ratio in 2015–17 after 40 years of follow up at age 37-54y by BMI and sex in the INCAP nutrition supplementation trial longitudinal cohort (n = 510 women, n = 346 men).**
(DOCX)

## Acknowledgments

**Disclaimer:** The findings and conclusions in this report are those of the authors and do not necessarily represent the official position of the Centers for Disease Control and Prevention.

## Author Contributions

**Conceptualization:** Nicole D. Ford, Reynaldo Martorell, Aryeh D. Stein.

**Formal analysis:** Nicole D. Ford.

**Funding acquisition:** Aryeh D. Stein.

**Investigation:** Reynaldo Martorell, Manuel Ramirez-Zea, Aryeh D. Stein.

**Methodology:** Nicole D. Ford, Neil K. Mehta, Cria G. Perrine, Aryeh D. Stein.

**Project administration:** Manuel Ramirez-Zea, Aryeh D. Stein.

**Supervision:** Reynaldo Martorell, Manuel Ramirez-Zea, Aryeh D. Stein.

**Visualization:** Nicole D. Ford.

**Writing – original draft:** Nicole D. Ford.

**Writing – review & editing:** Nicole D. Ford, Reynaldo Martorell, Neil K. Mehta, Cria G. Perrine, Manuel Ramirez-Zea, Aryeh D. Stein.

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
