## [Decision Letter · Decision Letter 0]

9 Apr 2020

PONE-D-20-00123

Lifecourse body mass index trajectories and cardio-metabolic disease risk in Guatemalan adults

PLOS ONE

Dear Dr. Ford,

Thank you for submitting your manuscript to PLOS ONE. After careful consideration, we feel that it has merit but does not fully meet PLOS ONE’s publication criteria as it currently stands. Therefore, we invite you to submit a revised version of the manuscript that addresses the points raised during the review process.

We would appreciate receiving your revised manuscript by May 24 2020 11:59PM. To enhance the reproducibility of your results, we recommend that if applicable you deposit your laboratory protocols in protocols.io, where a protocol can be assigned its own identifier (DOI) such that it can be cited independently in the future. For instructions see: http://journals.plos.org/plosone/s/submission-guidelines#loc-laboratory-protocols

We look forward to receiving your revised manuscript.

Kind regards,

Andreas Beyerlein

Academic Editor

PLOS ONE

Additional Editor Comments (if provided):

I understand that the data cannot be made available, but I would encourage the authors to make their analysis code publicly available, e.g. in an online archive such as OSF.

Reviewers' comments:

Reviewer's Responses to Questions

**Comments to the Author**

1. Is the manuscript technically sound, and do the data support the conclusions?

Reviewer #1: Partly

Reviewer #2: Partly

2. Has the statistical analysis been performed appropriately and rigorously? 

Reviewer #1: I Don't Know

Reviewer #2: Yes

3. Have the authors made all data underlying the findings in their manuscript fully available?

Reviewer #1: Yes

Reviewer #2: Yes

4. Is the manuscript presented in an intelligible fashion and written in standard English?

Reviewer #1: Yes

Reviewer #2: Yes

5. Review Comments to the Author

Reviewer #1: This is a well written manuscript, but there are some weaknesses in the analytical approach and discussion of the results. The paper is not giving entirely novel information, but the authors may be able to strengthen existing knowledge by expanding the paper with more detailed discussion of results. Especially, the rather one-sided sample composition with a very high rate of overweight and obese participants lacks appropriate discussion and meaningful conclusions.

There are some inconsistencies in spacing and formatting:

- l.168: “≥ 3 days of vigorous activity of ≥20 min/day”

- l.54 and l.64: inconsistent use of hyphen: “middle-income country” and “high income country”

- l.70-71: inconsistent use of dash and hyphen: “early life – especially (…) economies - have”, or l.68: (16-18) vs. l.69 (19–22).

Abstract:

- l.25: Please give the total N of study participants

- Please clarify that the three latent classes for men were collapsed to two classes for analysis.

Methods:

- l.82-84: What kind of nutritional intervention and at what age where infant recruited? Did the intervention influence any of the results/adjustment for intervention groups needed? What was the total n of participants in the study?

- l.90-92: Please provide details when informed consent was given by whom. As participants were recruited already in infancy, initial consent was given by the parents? Was an informed consent obtained from participants themselves at a later follow-up?

- Sentence in l. 99 to 101 is missing a verb: …trajectories were ?derived/estimated? from…

- Please revise the order of sub-headings/-sections of this section, e.g. the description of BMI trajectories should come after the description of measurements and probably after the statistical analysis section. Adding the trajectories part to the statistical analysis section is also possible

- There should be more information on actual numbers of BMI values involved in LCGA and a more detailed method description making the manuscript more readable without ref. 24.

- There is no actual reason to summarize the medium and high class from the LCGA modelling in men into one high class. To distinguish between severely obese and others would make sense.

- Please add a definition of all parameters included in the definition used for the CMD risk profiles, before the description of methods used to measure these parameters

- Same for the MetS: a short rationale would be helpful, as this syndrome is first mentioned in the methods section without context

- l.148: Change defines to defined

- l. 162: Please give a short rationale or reference for this dichotomization

- l.162: Was parity included in all models? Please specify

- l. 177-186: more fitting into the results section (together with Fig 2)

- l. 164: In what model were these two variables used? Can’t find them in the statistical analysis section. Please also provide a short rationale/reference on why to include them into the analysis

- l.201-202: What about the other lifestyle covariates (parity, multivitamin and alcohol use)? How were the covariates selected for the model?

- Please clarify for each model what is the exact outcome and what is the exact main predictor.

- A sensitivity analysis including all three latent classes for men might give some interesting results (depending on the size of the groups and outcome-diversity within these groups)

- Did you check for direct associations of BMI with different age? This might be more interesting than the usage of BMI classes.

Results:

- l. 256-261: are these numbers normal? They appear very high to me. Possibly discuss this in the discussion section

- l. 269: Clarify: obesity at what age/time point; obesity by BMI? What is the reason to regress Obesity defined by BMI on a BMI trajectory class.

Discussion:

- l. 329-334 and l.348-349: this indicates that the composition of this sample might not be optimal to answer the research question asked in the beginning of the manuscript. Please elaborate this aspect more

- l. l. 334: why? This conclusion seems kind of far fetched as most of the associations of BMI with CMD factors are no influence by the “BMI throughout life” i.e. the trajectories in this case but the present BMI.

- l. 337: Please define “large” differences in BMI trajectories.

- l.338: Why “pre-conceptual factors”? I think it is not possible to draw conclusions about this period based on the results.

- l. 341-349: Please clarify what your study adds to the current knowledge. The first and last sentence of this paragraph have no connection for me

- l. 350-351: in this study or in general?

- l. 350-362: in this paragraph you greatly highly the shortfalls of the sample composition in relation to the research question. Please elaborate, what insights you did get despite this unfavourable sample composition.

- l. 377-378: Measurements to build such trajectories of waist circumference or body fat were not available in your study? Or what is the reason to only built BMI trajectories in this cohort

- l.378: Please discuss more the role of early life undernutrition in your results

- l. 400: “and preventing weight gain in adulthood” I don’t think your research support this conclusion. Please rewrite the conclusion with more relation to the initial research question you formulated

Reviewer #2: Reviewer Comments to Authors

In this report, the authors evaluated the association of latent class trajectories of BMI from infancy to middle adulthood with cardiometabolic disease risk factors in a retrospective analysis of a cohort of participants in Guatemala enrolled in 1962-77 and followed through 2015-17. No associations between BMI latent class and cardiometabolic disease risk factors were found after adjustment for current BMI in men, and an association was found only between BMI latent class and obesity by BMI in women.

A number of comments and suggestions are offered:

1. Introduction: Greater contextualization of cardiometabolic disease in Latin America is suggested. It would be helpful to know where in the epidemiologic transition Guatemala lies. E.g., lines 45-46: what is the quantitative proportion of CMD disease burden in Latin America? Line 50: What is the prevalence of obesity in children and adults in Guatemala?

2. Introduction: A stronger narrative describing the connection between undernutrition and adult CMD is suggested. If “early life undernutrition is thought to increase risk of adult cardio-metabolic disease – particularly among people later exposed to hyper-caloric environments” (lines 52-53), is it the hyper-caloric environment that is the culprit? Additionally, more exposition on the relationship between childhood undernutrition and BMI trajectories would help support the rationale for this study.

3. Introduction, paragraph 3 (lines 56-63): For a reader who may be less familiar with methods, it would be helpful to more explicitly state why the reasons stated in this paragraph make understanding the role of younger-age body size on adult disease more challenging.

4. Introduction, paragraph 3 (lines 56-63): A statement establishing a stronger connection between the first sentence (starting line 56) and this second one (starting line 60) is needed: in what way does LCGA help understand the role of childhood body size on adult disease risk? The connection between the two points is not clear.

5. Methods: Can the authors provide a brief summary of the findings of the trial? It would be helpful context in this narrative to know if the intervention resulted in changes in physical development, and how (if at all) the supplementation changes the representativeness of this population to the remainder of the childhood population in Guatemala?

6. Methods (lines 110-114): Visually (Figure 1), the latent class BMI trajectories within-sex appear follow the same pattern. The primary difference appears to be that BMI levels are higher in the ‘high’ or ‘high/medium’ classes. In the context of “high levels of adiposity in both trajectory classes” (line 330), what is the authors’ interpretation of the populations the latent classes represent? In addition, what is the authors’ assessment of how clinically different/distinguishable the latent class trajectories are? Do the latent class trajectories represent a different population? The clinical difference between the latent class trajectories are not clear but should be discussed.

7. The authors state “Because BMI latent class trajectory class was not associated CMD risk after controlling for current BMI and given that early life BMI tracked into adulthood, our findings highlight the importance of healthy growth early in life and preventing weight gain in adulthood.” (Discussion, lines 398-400). Was a formal mediation analysis done? This conclusion commenting on how the data highlight the importance of “healthy growth early in life and preventing weight gain,” while consistent with a broad range of other studies, seems to reach beyond what the results of the observed associations between latent class BMI trajectories and CMD risk factors show.

8. Given the generally high level of adiposity in all participants, have the authors considered further examining the subset of participants (if any) who are “metabolically normal obese”?

9. Minor comment: Though this may be semantic, there is a clinical difference between CMD, CMD risk factors (e.g. obesity, cholesterol levels), and CMD risk (e.g., in the context of risk estimation models like the pooled cohort equations that are widely used in clinical practice for primary prevention of cardiovascular disease). The independent variables in this analysis seems to be CMD risk factors, so the authors may consider using “CMD risk factors” rather than “CMD risk” throughout to describe their results.

6. PLOS authors have the option to publish the peer review history of their article (what does this mean?). If published, this will include your full peer review and any attached files.

Reviewer #1: No

Reviewer #2: No

---

## [Author Response · Author response to Decision Letter 0]

18 Aug 2020

We have responded to the reviewer comments in the attached file titled "Responses to Reviewers."

---

## [Decision Letter · Decision Letter 1]

11 Sep 2020

PONE-D-20-00123R1

Lifecourse body mass index trajectories and cardio-metabolic disease risk in Guatemalan adults

PLOS ONE

Dear Dr. Ford,

Thank you for submitting your manuscript to PLOS ONE. After careful consideration, we feel that it has merit but does not fully meet PLOS ONE’s publication criteria as it currently stands. Therefore, we invite you to submit a revised version of the manuscript that addresses the points raised during the review process.

We look forward to receiving your revised manuscript.

Kind regards,

Andreas Beyerlein

Academic Editor

PLOS ONE

Additional Editor Comments (if provided):

As mentioned in my previous response on this paper, I understand that the authors may not be able to make the data publicly available - but I cannot see any reason why this would not be possible with respect to the analysis code. This would be very much in line with reproducible and open science.

Reviewers' comments:

Reviewer's Responses to Questions

**Comments to the Author**

1. If the authors have adequately addressed your comments raised in a previous round of review and you feel that this manuscript is now acceptable for publication, you may indicate that here to bypass the “Comments to the Author” section, enter your conflict of interest statement in the “Confidential to Editor” section, and submit your "Accept" recommendation.

Reviewer #1: All comments have been addressed

Reviewer #2: All comments have been addressed

2. Is the manuscript technically sound, and do the data support the conclusions?

Reviewer #1: Yes

Reviewer #2: Yes

3. Has the statistical analysis been performed appropriately and rigorously? 

Reviewer #1: Yes

Reviewer #2: Yes

4. Have the authors made all data underlying the findings in their manuscript fully available?

Reviewer #1: Yes

Reviewer #2: Yes

5. Is the manuscript presented in an intelligible fashion and written in standard English?

Reviewer #1: Yes

Reviewer #2: Yes

6. Review Comments to the Author

Reviewer #1: (No Response)

Reviewer #2: (No Response)

7. PLOS authors have the option to publish the peer review history of their article (what does this mean?). If published, this will include your full peer review and any attached files.

Reviewer #1: No

Reviewer #2: No

---

## [Author Response · Author response to Decision Letter 1]

23 Sep 2020

We have uploaded the code to OSF. Please see:

Ford N. SAS code for “Lifecourse body mass index trajectories and cardio-metabolic disease risk in Guatemalan adults.” [Internet]. OSF; 2020. Available from: osf.io/5cvw8

---

## [Editor Report · Decision Letter 2]

28 Sep 2020

PONE-D-20-00123R2

Lifecourse body mass index trajectories and cardio-metabolic disease risk in Guatemalan adults

PLOS ONE

Dear Dr. Ford,

Thank you for submitting your manuscript to PLOS ONE. After careful consideration, we feel that it has merit but does not fully meet PLOS ONE’s publication criteria as it currently stands. Therefore, we invite you to submit a revised version of the manuscript that addresses the points raised during the review process.

We look forward to receiving your revised manuscript.

Kind regards,

Andreas Beyerlein

Academic Editor

PLOS ONE

Additional Editor Comments (if provided):

Please add the URL where the SAS code can be found to the end of the Methods section. After this, the manuscript can be accepted.

---

## [Author Response · Author response to Decision Letter 2]

28 Sep 2020

We added the following sentence to the end of the Methods section:

“SAS code is available online via OSF at: https://osf.io/5cvw8/.”

---

## [Editor Report · Decision Letter 3]

6 Oct 2020

Lifecourse body mass index trajectories and cardio-metabolic disease risk in Guatemalan adults

PONE-D-20-00123R3

Dear Dr. Ford,

We’re pleased to inform you that your manuscript has been judged scientifically suitable for publication and will be formally accepted for publication once it meets all outstanding technical requirements.

Kind regards,

Andreas Beyerlein

Academic Editor

PLOS ONE
---

## [Editor Report · Acceptance letter]

13 Oct 2020

PONE-D-20-00123R3 

Lifecourse body mass index trajectories and cardio-metabolic disease risk in Guatemalan adults 

Dear Dr. Ford:

I'm pleased to inform you that your manuscript has been deemed suitable for publication in PLOS ONE. Congratulations! Your manuscript is now with our production department. 

Kind regards, 

on behalf of

Dr. Andreas Beyerlein 

Academic Editor

PLOS ONE